# Receptor-Mediated AKT/PI3K Signalling and Behavioural Alterations in Zebrafish Larvae Reveal Association between Schizophrenia and Opioid Use Disorder

**DOI:** 10.3390/ijms23094715

**Published:** 2022-04-25

**Authors:** Siroshini K. Thiagarajan, Siew Ying Mok, Satoshi Ogawa, Ishwar S. Parhar, Pek Yee Tang

**Affiliations:** 1Department of Mechatronics and Biomedical Engineering, Universiti Tunku Abdul Rahman, Kajang 43000, Malaysia; t.siroshini@gmail.com (S.K.T.); moksy@utar.edu.my (S.Y.M.); 2Jeffery Cheah School of Medicine and Health Sciences, Monash University Malaysia, Subang Jaya 47500, Malaysia; satoshi.ogawa@monash.edu (S.O.); ishwar@monash.edu (I.S.P.)

**Keywords:** addiction, schizophrenia, zebrafish, behaviour, dopamine

## Abstract

The link between substance abuse and the development of schizophrenia remains elusive. In this study, we assessed the molecular and behavioural alterations associated with schizophrenia, opioid addiction, and opioid withdrawal using zebrafish as a biological model. Larvae of 2 days post fertilization (dpf) were exposed to domperidone (DMP), a dopamine-D2 dopamine D2 receptor antagonist, and morphine for 3 days and 10 days, respectively. MK801, an N-methyl-D-aspartate (NMDA) receptor antagonist, served as a positive control to mimic schizophrenia-like behaviour. The withdrawal syndrome was assessed 5 days after the termination of morphine treatment. The expressions of schizophrenia susceptibility genes, i.e., *pi3k*, *akt1*, *slc6a4*, *creb1* and *adamts2*, in brains were quantified, and the levels of whole-body cyclic adenosine monophosphate (cAMP), serotonin and cortisol were measured. The aggressiveness of larvae was observed using the mirror biting test. After the short-term treatment with DMP and morphine, all studied genes were not differentially expressed. As for the long-term exposure, *akt1* was downregulated by DMP and morphine. Downregulation of *pi3k* and *slc6a4* was observed in the morphine-treated larvae, whereas *creb1* and *adamts2* were upregulated by DMP. The levels of cAMP and cortisol were elevated after 3 days, whereas significant increases were observed in all of the biochemical tests after 10 days. Compared to controls, increased aggression was observed in the DMP-, but not morphine-, treated group. These two groups showed reduction in aggressiveness when drug exposure was prolonged. Both the short- and long-term morphine withdrawal groups showed downregulation in all genes examined except *creb1*, suggesting dysregulated reward circuitry function. These results suggest that biochemical and behavioural alterations in schizophrenia-like symptoms and opioid dependence could be controlled by common mechanisms.

## 1. Introduction

Schizophrenia is a neuropsychiatric disorder affecting roughly 1% of the entire population, with debilitating effects on people’s lives [1]. This disorder can be distinguished by the presentation of two classes of symptoms, namely the positive and negative symptoms [2]. The positive symptoms comprise delusions, hallucinations, and bizarre thoughts, while the negative symptoms include apathy, blunted affect, decreased motivation, and social withdrawal. Similarly to most psychiatric disorders, schizophrenia is complex in its origins. The co-occurring substance use disorder has been highly linked to schizophrenia, having been related with great risk of illness and injuries [3]. Cannabis use has been associated with schizophrenia, and a history of substance use was more likely to increase aggressive behaviour in male patients [4]. Schizophrenia patients and substance abusers often showed similarity in positive symptoms, negative symptoms, general psychopathology symptoms or anxiety [5]. Apart from the impaired cognitive functioning, substance misuse is associated with poorer outcomes in psychosis [3]. Studies have also associated substance use disorder with an earlier onset of schizophrenia [6].

Recurrent long-term exposure to drugs produces withdrawal symptoms when their use is discontinued, due to the filtering of drugs from the body system. After the user has built up a tolerance to the substance, the body’s homeostatic counter adaptations will result in a compound phenomenon. The withdrawal symptoms and the severity of the symptoms depend on tolerance and types of drugs, and may vary from patient to patient [7]. Clinical symptoms associated with withdrawal include headache, aches, anxiety, hallucinations, seizure, or influenza-like syndromes [8]. Although the appearance of psychosis is not common for withdrawal, it was found for cocaine, cannabis, opiates, and amphetamines [9,10,11,12,13,14]. Furthermore, the desire to avoid and escape from the adverse symptoms of withdrawal can unconsciously lead to drug seeking, causing relapse [15].

Dopamine (DA) is a well-known neurotransmitter that is involved in the neurobiology and symptoms of numerous psychiatric diseases, including schizophrenia. In addition, it is crucial in the brain reward system and associated with drug abuse potential [16,17]. Early studies revealed that DA receptors were involved in the therapy or etiology of psychiatric disorders [18,19]. The action of DA is mediated through binding to two types of receptors, the D1 and D2 receptors [20]. Binding of DA to its receptors induces several second messengers, especially cyclic adenosine monophosphate (cAMP) [21]. The dopamine D2 receptor (DRD2), a crucial element in the dopaminergic system, is one of the direct targets in treating schizophrenia. When DA binds to the Gαi/o-coupled DRD2, adenylate cyclase activity is inhibited and the production of the second messenger, cAMP, is suppressed [20,22]. The cAMP-dependent signal transduction is negatively regulated by serotonin, 5-hydroxytryptamine (5-HT). Serotonin transporter (5-HTT) encoded by the solute carrier family 6 member 4 (*slc6a4*), a serotonin transporter gene, regulates the 5-HT reuptake in presynaptic neurons [23].

The phosphoinositide 3-kinases (PI3K)–AKT (v-akt murine thymoma viral oncogene homolog, also known as protein kinase B) signalling pathway is an important downstream intracellular signalling pathway of DRD2 [24], and PI3K–AKT signalling is the critical factor for neuronal survival [25,26]. AKT, a downstream target of PI3K, plays an important role in the signal transduction of the PI3K–AKT pathway. Phosphatidylinositol-3, 4, 5-triphosphate (PIP3) leads to the membrane recruitment of AKT along with phosphoinositide-dependent protein kinase-1 (PDK1). AKT is then phosphorylated by PDK1 via activation of the Thr308 site in the T-loop [27]. AKT signalling will further activate cAMP responsive element binding protein 1 (CREB1), a leucine zipper transcription factor [28]. Dopaminergic signalling regulates the expression of a disintegrin and metalloproteinase with thrombospondin motifs 2 (ADAMST2) through cAMP/CREB signalling [29]. These downstream targets of cytoplasmic PI3K are implicated in a variety of cellular processes such as cell apoptosis, cell metabolism and intracellular vesicle transport [30].

Based on the “primary addiction hypothesis” [31] or the “reward deficiency syndrome” [32], both schizophrenia and substance use disorders share a common pathophysiology in overlapping neural circuits. Thus, substance use may produce dysfunction in the brain reward circuit that presents in patients with schizophrenia. Indeed, the PI3K–AKT signalling pathway may be involved in the susceptibility to developing schizophrenia and a co-occurring substance use disorder. Studies have showed that this signalling pathway is related to psychiatric diseases such as schizophrenia and depression [33,34,35]. Accumulating evidence also supports the role of the PI3K–AKT activity in binge drinking [36], cocaine-seeking behaviour [37] and methamphetamine-induced cell apoptosis [38]. However, to the best of our knowledge, no study to date has compared the effects of DRD2 and morphine (mu) receptor antagonism on the PI3K–AKT signalling pathway.

Zebrafish have widely emerged as an important vertebrate system for psychiatric disorders and substance abuse. To date, various behavioural assays of both larval and adult zebrafish have been developed [39,40,41]. The drug-responsive “reward” or reinforcement pathway in zebrafish makes it a useful model in the investigation of the mechanisms underlying drug addiction [5,6]. The zebrafish genome has significant homology to human genome, suggesting genes linked to psychiatric disorders can be studied [42]. Since zebrafish is well-suited for molecular and genetic analysis of temporal and spatial gene expression [43], the basic function of these genes can be explored during neural development.

In animal models such as rodents, the most common pharmacological strategy to elicit schizophrenia-like behaviors is based on the inhibition of N-methyl-D-aspartate (NMDA) receptor [44] using MK801, a non-competitive NMDA receptor antagonist [45]. This drug is able to mimic the hypofunction of NMDA receptor [46], which leads to behavioural effects that resemble the full spectrum of positive, negative, and cognitive symptoms of schizophrenia [47,48]. These key outcomes induced by MK801 were replicated in zebrafish, supporting previous studies and the use of zebrafish to study schizophrenia-related endophenotypes [49]. In this study, we assessed changes in the gene expression of zebrafish larvae subjected to both short- and long-term exposure to DMP and morphine. In this way, we could verify whether the gene expression changes in the PI3K–AKT signalling pathway will lead to biochemical and behavioural changes associated with schizophrenia, opioid addiction, and opioid withdrawal.

## 2. Results

### 2.1. Systemic Toxicity

A systemic toxicity test was conducted to identify the adverse effects that appeared after exposure to DMP and morphine. Table A1 depicts the highest concentrations of each drug in which no phenotypic abnormalities, including abnormal heart rate, death, malformation and scoliosis, were observed at the development stages tested. The selected maximum tolerated concentrations (MTCs) for DMP and morphine were 3.13 µM and 0.80 µM, respectively (Figure 1A).

### 2.2. Startle Habituation Response

Areas under the curve (AUCs) were established by monitoring the movement of larvae during their exposure to tap stimuli. DMP- and morphine-treated groups demonstrated a higher AUC compared to the control (Appendix A, F (5,75) = 16.399, *p* = 0.001), with the most significant difference in 3.13 μM DMP (*p* = 0.05) and 0.8 μM morphine (*p* = 0.05).

### 2.3. Effects of Drugs on Gene Expression

The gene expression study was conducted in order to compare and observe the changes, corroborating the observed behavioural abnormalities. A 3-day exposure to MK801, DMP and morphine resulted in a significant increase in the expression of *adamts2* (Figure 2A). When the treatment was prolonged to 10 days, different gene expression patterns were observed. In the MK801-treated group, all genes except *akt1* were significantly upregulated (Figure 2B), while *akt1* was significantly reduced. Treatment with DMP significantly downregulated *akt1*, while *creb1* and *adamts2* were upregulated (Figure 2B). Exposure to morphine significantly downregulated *akt1*, *pi3k* and *slc6a4* expression, but there was no effect on *creb1* and *adamts2* (Figure 2B).

We also examined the effect of morphine withdrawal treatment on gene expression. After the short-term drug exposure followed by 5-day morphine withdrawal, *akt1*, *pi3k* and *slc6a4* gene expression were significantly downregulated, while *creb1* was significantly increased (Figure 2C). Long-term morphine exposure followed by 5-day withdrawal suppressed expression of *akt1*, *pi3k* and *adamts2*, while the expression of *creb1* was upregulated (Figure 2D).

### 2.4. Effects of Drugs on cAMP, Cortisol and Serotonin-Based Signalling

Biochemical experiments were performed to elicit alterations of cAMP, cortisol and serotonin and further provide insight into the effects of drugs on related pathways. Three-and five-day exposure with MK801, DMP and morphine significantly increased cAMP (Figure 3A,B) and cortisol (Figure 4A,B) levels. Similarly, cAMP (Figure 3C,D) and cortisol (Figure 4C,D) levels were significantly increased after short- and long-term morphine exposure followed by 5-day withdrawal. On the other hand, serotonin levels were increased in larvae after 3-day exposure to MK801 and DMP, but morphine had no effect (Figure 5A). Under the longer exposure regime, treatment with DMP and morphine had a significant effect on serotonin levels, but there was no effect in the MK801-treated group (Figure 5B). In the morphine withdrawal groups, serotonin levels were significantly increased (Figure 5C,D).

### 2.5. Effects of Drug Exposure and Morphine Withdrawal on Mirror Biting Behaviour

To evaluate the zebrafish aggressiveness level, a mirror biting assay was conducted. Larvae treated with MK801 for short- and long-term periods exhibited significantly higher mirror biting scores as compared to other groups (Figure 6A,B). Short-term exposure to DMP also induced a significant increase in mirror biting behaviour (Figure 6A), while long-term exposure to DMP suppressed the mirror biting score (Figure 6B). In morphine-treated groups, the frequency of mirror biting was only reduced in the long-term exposed group (Figure 6B). In the morphine-withdrawn larvae, the mirror biting behaviour was significantly decreased (Figure 6C,D).

## 3. Discussion

With the MTC of each drug, zebrafish larvae of 2 dpf were treated for 3 days (short-term = 5 dpf) and 10 days (long-term = 12 dpf). A 5-day morphine withdrawal was applied to larvae followed by 3-day (=10 dpf) and 10-day (=17 dpf) exposure to morphine. Two dpf larvae were chosen because this is the key stage of brain ontogeny, including for monoaminergic neurons in zebrafish [50]. The DA system in zebrafish is well-characterized and is fully developed by 4 dpf [40,51,52]. Moreover, in zebrafish larvae, features of the blood–brain barrier appear at 3 dpf, and are fully developed by 10 dpf [53]. Hence, the underdeveloped blood–brain barrier at 2 dpf is less likely to prevent exposed drugs, in particular DMP [54], from entering the central nervous system. However, the efficacy of exposure to DMP in later stages might have compromised due to the developed blood–brain barrier.

Startle response, which is rapidly produced after the initiation of a sudden or strong stimulus [55,56], reflects the activation of motor tracts in the brainstem, particularly the bulbopontine reticular formation [57]. This primitive reflex displays several forms of plasticity such as prepulse inhibition (PPI) and habituation, which play a key role in filtering relevant and irrelevant sensory information [58]. A decline in PPI in schizophrenia patients has been consistently observed [59,60,61,62,63,64], and it has thus been proposed as a biomarker of the disease [65,66]. Although PPI in zebrafish is remarkably less complex than in rodents, it can be disrupted in genetic, pharmacological and other models relevant for schizophrenia, indicating that the mechanism involved in sensorimotor gating is evolutionary conserved [67]. Therefore, the zebrafish model is applicable in studying basic mechanisms of sensorimotor gating and its association with schizophrenia [68]. Moreover, we demonstrated that tap stimuli could significantly impair the habituation response in the MK801-induced schizophrenia model of zebrafish [69]. In our dose optimization, 3.13 μM DMP and 0.8 μM morphine were both shown to elicit the most impaired habituation response in zebrafish larvae. Therefore, these two doses were selected as the optimal doses for the subsequent drug treatment.

The pathogenesis of schizophrenia is supported by the dysregulation of the signalling related to neurotransmitters, intracellular signal transduction, and neural development [70]. Larvae treated with DMP and morphine showed a certain degree of similarity with MK801 in alteration of *akt1* and *adamts2*. Prolonged blockade of NMDA receptors by MK801 inhibits DRD2 function [71], while chronic morphine exposure leads to a decrease in the level of DRD2 mRNA [72]. The reduction in *akt1* expression in DMP- and morphine-exposed larvae could be due to the long-term inhibition of DRD2. It may also be explained by the inhibition of cell apoptosis by the cell survival factor through activating a specific signalling pathway, as well as the PI3K–AKT pathway. It has been shown that transfection of constitutively active AKT prevents cell apoptosis while a dominant negative AKT induces cell apoptosis [73]. In addition, inhibitors of the PI3K–AKT pathway can sensitize cells to apoptotic stimuli by activation of opioid receptors [73,74]. AKT is known to act as a negative regulator of the mitochondrial protein, GSK3 [75], which has been implicated in the development of schizophrenia [76]. In neurons derived from schizophrenia patients, the sensitivity to PI3K/GSK3 signaling inhibition is decreased [77]. AKT activity has been shown to decrease in certain brain regions of patients with major depressive disorder and schizophrenia [78]. An endogenous neuro-steroid in the central nervous system, pregnenolone, was indicated to normalize schizophrenia-like behaviours via the AKT signalling [33].

Our data suggest that there was an activation of cAMP after all treatments. The cAMP signalling pathway is a part of the drug addiction-related pathways and has been implicated in neurological diseases including drug addiction and schizophrenia [79]. Chronic morphine treatment induces upregulation of the cAMP signalling pathway in many brain areas such as the ventral tegmental area and the nucleus accumbens, which are critical either for the reinforcing effects of drugs or for the development of somatic signs of opioid withdrawal [80]. The activated cAMP increases neurotransmitter release in the central glutamatergic and GABAergic synapses [81] and, therefore, enhances synaptic plasticity [82]. As DRD2 mediates the inhibition of intracellular cAMP formation [83], the blockage of the DRD2 receptors by DMP may have led to activation of cAMP.

CREB is an intracellular protein that regulates the expression of genes that are important in dopaminergic neurons. In patients with schizophrenia, there are some risk variants in the promoter region of *creb* [35]. In this study, *creb* expression was significantly upregulated in the morphine withdrawal groups and could be induced by the increased cAMP levels. In addition, all treatments increased ADAMTS2 expression, which could also be due to the high cAMP production. The mRNA of *adamts2* and its protein are specifically found in mesolimbic and meso-cortical dopaminergic regions [29]. This gene is overexpressed in schizophrenia patients, and it has been implicated in the positive symptoms of the disease [84,85]. In the morphine withdrawal group (followed by 10-day exposure), *adamts2* mRNA levels were significantly reduced, which could be due to long-term inhibition of DRD2 by morphine. Our result was in-line in line with the finding by Ruso-Julve et al. [29] that haloperidol, a DRD2 antagonist, induced inhibition of *adamts2* expression.

A previous study reported that aggressive behaviour is associated with positive symptoms of schizophrenia [86]. In the present study, mirror biting behaviour was used to examine the effect of drug exposures on aggressiveness in zebrafish larvae [87]. In this study, different durations of DMP and morphine exposure caused reversed results for the same drug. Mirror-biting frequency was increased in both groups treated with short- and long-term exposure to MK801, when compared to the control. This could be due to the antagonistic effect of MK801 on NMDA receptor and its associated activation of the dopaminergic activity [88,89,90]. On the other hand, a previous study in adult (6–8 months old) zebrafish demonstrated that acute (15-min) treatment with MK801 induced a decrease in aggressive behaviour [91]. Similarly, several studies have shown that MK801 induces deficits in the social interaction parameters in rodents and zebrafish [92,93].

Both DMP- and morphine-treated larvae showed a significant reduction in mirror-biting frequency after long-term drug exposure. The effect of DMP on aggression has also been demonstrated in cats but not in rats [94,95]. As DMP is a peripheral DRD2 antagonist, the effect of DMP on aggression remains debatable. Alternatively, the suppressive effect of DMP on mirror biting behaviour could be partially due to its motor impairment effect [94,96]. Similar to our findings, acute administration of opiates reduced aggressive behaviours in human and mice [97,98,99]. The suppressive effects of morphine and other opioids on aggressive behaviours have been reported in various animals [100,101]. The anti-aggressive effect of morphine is also considered as a part of the opiate sedative and tranquilizing effects, which might have affected expressions of aggressiveness [97,98,102]. We also found significantly reduced aggression in zebrafish larvae that experienced morphine withdrawal. On the other hand, in mice, the removal of morphine led to increases in aggression such as attack bites and threats [103,104]. Although morphine withdrawal aggression has not been reported in zebrafish, morphine withdrawal triggers withdrawal-like states and induces strong anxiogenic behaviours such as decreased exploratory behaviour and increased erratic movement [105,106]. Our results also showed that long-term exposure to morphine and its withdrawal significantly increased whole-body cortisol levels. Increased zebrafish whole-body cortisol secretion has been associated with the anxiogenic effect of withdrawal from several addictive substances, including ethanol and morphine [107,108]. Hence, the reduction in aggressiveness in the morphine withdrawal group could be a result of reduced general locomotion due to the anxiogenic effect.

We also examined the effect of drug treatments on the stress–response pathway by measuring the stress hormone cortisol, as it is crucial in triggering the expression of vulnerability to mental disorders. There was an increase in whole-body cortisol production in all treated larvae compared to the controls. Studies have shown elevated cortisol in first episode and recent-onset psychotic patients, and increased activity of systemic cortisol metabolism in schizophrenia patients [87,88]. Like mammals, in teleosts, the release of cortisol results from a series of hormonally regulated events involving the activation of the hypothalamus–pituitary–interrenal (adrenal homolog of teleost fish) axis with corticotropin-releasing factor (CRF) being the initial cascade hormone. CRF-related peptides act within the dorsal raphe nucleus, the major source of serotonin in the brain to alter the neuronal activity of specific subsets of serotonergic neurons and to influence stress-related behaviour [109,110,111,112]. Moreover, after long-term exposure, the increase in serotonin levels found in the morphine-treated group might be aggravated by the downregulation of *slc6a4*. There is evidence that lower *slc6a4* transcriptional levels due to the short allele in its upstream region result in reduced reuptake of serotonin [113,114]. The elevated serotonin levels in the long-term DMP- and morphine-treated groups and withdrawal groups might reflect their anxiogenic condition.

## 4. Materials and Methods

### 4.1. Zebrafish Strains and Housing Conditions

Wild-type zebrafish embryos at 0 h post fertilization (hpf) were collected from Danio Assay Laboratories Pvt. Ltd., Universiti Putra Malaysia. The embryos were maintained at 27 °C in embryo medium (5 mM NaCl, 0.17 mM potassium chloride (KCl), 0.33 mM calcium chloride (CaCl_2_), and 0.33 mM magnesium sulfate (MgSO_4_), pH 7.4). Unfertilized, unhealthy, and dead embryos were removed. The hatched larvae were fed starting 7 days post fertilization (dpf) with live paramecium. All experiments were performed with the approval of animal ethics by Universiti Tunku Abdul Rahman Research Ethics and Code of Conduct (U/SERC/18/2020). The health of the zebrafish larvae was continually monitored by occasionally visualizing the mortality, heartbeat, and locomotor activity of the larvae within the time frame of experimental procedures.

### 4.2. Optimization of Drug Dosages

To determine the suitable concentrations for testing, we performed optimization using a concentration series of drugs (0.2 to 50.0 µM). The MTC of each drug was identified based on heart rate observations under a dissecting microscope, the survival rate, and the deleterious phenotypic effects such as hatching rate, scoliosis rate, and heart rate appearing on the larvae after incubation with the drugs until the embryos reached 10 dpf [115]. Morphine sulfate pentahydrate (Lipomed AG, Switzerland) was dissolved in distilled water, while domperidone (DMP) (Sigma Aldrich) was dissolved in dimethyl sulfoxide (DMSO). All compounds were administered into individual wells of 96-well microtiter plates with 100 µL of working treatment solution, where the final concentration of DMSO was less than 0.01%. Zebrafish embryos at 24 hpf were exposed to different concentrations of drugs. The treatment solutions were replenished daily until the end of treatment. We used n = 15 embryos (3 independent experiments, 5 embryos per experiment) for each drug concentration as well as for the controls.

In order to determine the efficacy of the selected MTCs in inducing schizophrenia-like behaviour, a startle habituation assay was conducted. Two groups (n = 15) of larvae were treated separately with five doses each of DMP or morphine from approximately 2 dpf to 5 dpf. A separate group of larvae were held in embryo medium to serve as control. At 6 dpf, each group of larvae was individually transferred to a petri dish that was filled with embryo medium and left to acclimatize for 30 min. Each larva was exposed to a series of 36 tap stimuli at 1 s intervals. The taps were produced by a 9 volt DC solenoid that was driven by an Arduino Uno microcontroller. A recording camera at 30 frames per second (fps) was positioned above the swim arena to capture the swimming activity of the larvae. Native video files were processed using Toxtrac, an open-source tracking software [116], and the total activity was determined by calculating the AUC.

### 4.3. Drug Treatments

The MTC of each drug (Figure 1A; Table A1) was identified in order to optimize the short-term (3 day) and long-term (10 day) treatment protocols (Figure 1B). Dizocilpine, also known as MK801 (Sigma Aldrich, St. Louis, MO, USA), an NMDA receptor antagonist that can mimic some aspects of schizophrenia, such as hyperlocomotor behaviour, depressive symptoms and cognitive deficits [117,118], served as a positive control. MK801 was dissolved in distilled water to achieve the final concentration of 5.0 µM. According to previous studies, this dosage was able to induce social interaction deficit, a behavioural feature observed in schizophrenia, in zebrafish [91,92]. Meanwhile, embryo medium served as vehicle control. Zebrafish embryos were randomly divided into control and three treatment groups. Three groups (n = 50/group) of embryos starting at 2 dpf were examined for the effect of short-term and long-term treatments. The withdrawal syndrome of the morphine-treated larvae was assessed 5 days after the termination of treatment.

### 4.4. RNA Extraction and Real-Time PCR

The larvae were euthanized by rapid cooling in ice water (0–4 °C) for at least 20 min [119]. Three independent groups of zebrafish larvae used for each treatment were dissected to extract the whole brains. The brains of each group (n = 50) were pooled for total RNA extraction using TRIzol reagent (Thermo Fisher Scientific, Waltham, MA, USA), based on the protocol of Chomczynski and Mackey [120]. The quality of RNA was assessed by gel electrophoresis, and the total RNA concentration was determined using a spectrophotometer. The double-stranded cDNA was synthesized using a Tetro cDNA Synthesis Kit (Bioline Reagents Ltd., London, UK). Five genes related to neuronal synaptic transmission (Figure 6C) were selected. Quantitative real-time PCR was performed with SensiFAST™ SYBR**^®^** No-ROX kit (Bioline Reagents Ltd., London, UK) to amplify *slc6a4*, *pi3k*, *akt1*, *creb1* and *adamts2* with beta actin (β-actin) as the housekeeping gene in a qTOWER3 G Real-Time PCR machine (Analytik Jena, Jena, Germany). The primer sequences were adapted from Cheng et al. [13] and Senay et al. [10] (Table A2). cDNA was diluted in series ranging from 100 ng/μL to 0.01 ng/μL to generate a standard curve. The relative expression of each gene was calculated from the average cycle threshold (CT) readings at each point of the standard curve using the formula E = 10^(1/slope)^. The relative expression levels of genes of interest were then normalized to the housekeeping gene.

### 4.5. Total Protein Purification

Zebrafish larvae were washed with cold phosphate buffered saline (PBS) and stored overnight at −20 °C. Cell lysis was conducted by two freeze–thaw cycles. The whole-body sample was homogenized in PBS containing 1% Triton-X and sonicated for 10 min. The homogenates were centrifuged at 5000× *g* for 5 min. The supernatant was collected, and subsequently the protein concentration was determined using Bradford assay. The purified protein was subjected to biochemical assays immediately.

### 4.6. Biochemical Assays

The concentrations of cAMP, serotonin and cortisol were determined using commercial enzyme-linked immunosorbent assay (ELISA) kits (Elabscience**^®^** and (Cusabio Biotech**^®^**, respectively) according to the manufacturer’s instructions. The optical density (OD value) of each well was determined at 450 nm for all three assays.

### 4.7. Behavioural Testing

The mirror biting test is a simple and effective method for analysing the aggressive behaviour of adult and larval zebrafish [121]. To assess the aggressiveness of larvae subjected to different days of treatment, a test tank of dimension 15.0 cm × 5.0 cm × 4.0 cm (length × width × height) with a mirror (5.0 cm × 4.0 cm) on the sidewall was filled with water (Figure 1D). A larva was left undisturbed to acclimate for 15 min. After acclimatization, the number of times that the larva bit the mirror image in 5 min was manually monitored and recorded.

### 4.8. Statistical Analysis

All experimental data obtained from short-and long-term exposure to drugs were expressed as mean ± SEM. The differences between samples were analysed using the one-way analysis of variance (ANOVA), followed by post-hoc Tukey’s test. The means for control and withdrawal groups were analysed using Student’s *t*-test. Statistical analysis was performed using Statistical Programme for Social Sciences (SPSS version 22). A *p*-value of less than 0.05 (*p* < 0.05) was considered statistically significant.

## 5. Conclusions

In conclusion, there is a possible association between schizophrenia and opioid addiction. Our results showed that the genes involved in the PI3K–AKT signalling pathway play important roles in the modulation of behavioural and neurological activity induced by DMP and morphine. Our findings could advance the understanding of the pathogenetic association between these two disorders.

## Figures and Tables

**Figure 1 ijms-23-04715-f001:**
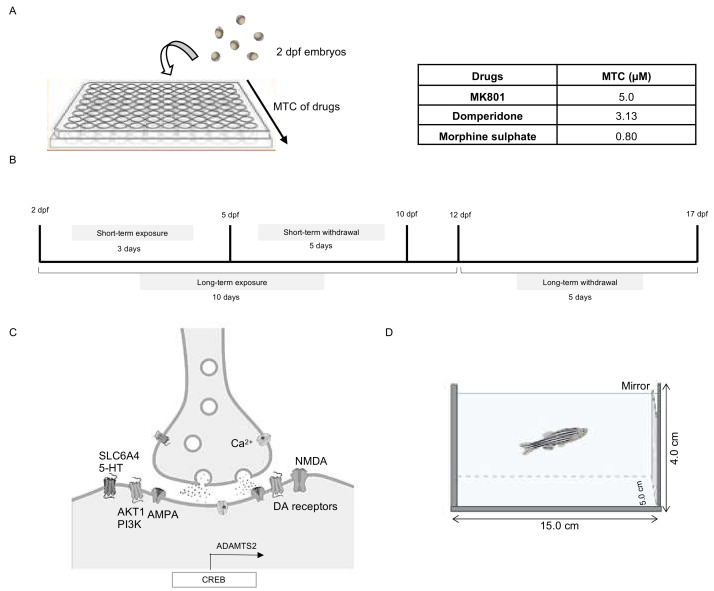
Experimental design schematics. (**A**) Embryos at 48 hpf were treated to obtain the MTC of the drugs. (**B**) Exposure timeline of the drugs. (**C**) Synaptic localization of all genes examined. (**D**) Aggression testing using a test tank with a mirror. ADAMTS2: a disintegrin and metalloproteinase with thrombospondin motifs 2, AKT1: v-akt murine thymoma viral oncogene homolog 1, AMPA: α-amino-3-hydroxy-5-methyl-4-isoxazolepropionic acid, CREB: cAMP responsive element binding protein, DA: dopamine, dpf: days post fertilization, hpf: hours post fertilization, MTC: maximum tolerated concentration, NMDA: N-methyl-D-aspartate, PI3K: phosphoinositide 3-kinase, SLC6A4: solute carrier family 6 member 4, 5-HT: 5-hydroxytryptamine.

**Figure 2 ijms-23-04715-f002:**
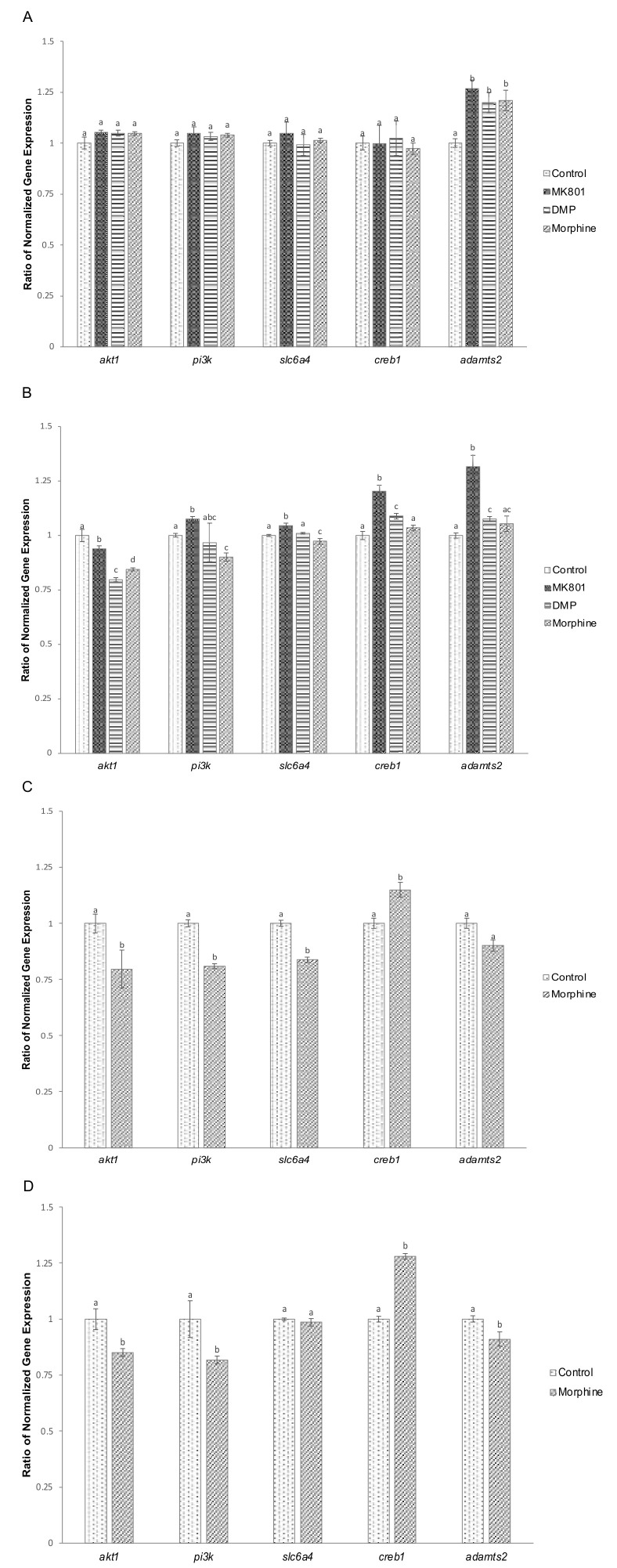
Altered gene expression in zebrafish larvae following exposure to MK801, DMP and morphine, and morphine withdrawal. (**A**) Effect of short-term (3-day) drug exposure on expression of *akt1* (F (3,32) = 1.386, *p* = 0.265), *pi3k* (F (3,32) = 1.127, *p* = 0.299), *slc64a* (F (3,32) = 1.721, *p* = 0.182), *creb1* (F (3,32) = 0.522, *p* = 0.667) and *adamst2* (F (3,32) = 2.771, *p* = 0.058). (**B**) Effect of long-term (10-day) drug exposure on expression of *akt1* (F (3,32) = 14.546, *p* < 0.001), *pi3k* (F (3,32) = 6.650, *p* < 0.001), *slc64a* (F (3,32) = 25.115, *p* < 0.001), *creb1* (F (3,32) = 29.607, *p* < 0.001) and *adamst2* (F (3,32) = 90.337, *p* < 0.001). (**C**) Effect of short-term morphine exposure followed by 5-day withdrawal on expression of *akt1* (*t* (16) = 8.945, *p* = 0.003), *pi3k* (*t* (16) = 8.082, *p* = 0.001), *slc6a4* (*t* (16) = 5.591, *p* = 0.001), *creb1* (*t* (16) = 6.394, *p* = 0.001), *adamts2* (*t* (16) = 0.306, *p* = 0.064). (**D**) Effect of long-term morphine exposure followed by 5-day withdrawal on expression of *akt1* (*t* (16) = 8.945, *p* = 0.003), *pi3k* (*t* (16) = 27.493, *p* < 0.001), *slc6a4* (*t* (16) = 2.445, *p* = 0.058), *creb1* (*t* (16) = 6.102, *p* < 0.001), *adamts2* (*t* (16) = 5.709, *p* = 0.001). Data expressed as mean ± SEM, *n* = 50 of 3 independent experiments. Different lowercase letters indicate statistically significant values, *p* < 0.05. DMP: domperidone.

**Figure 3 ijms-23-04715-f003:**
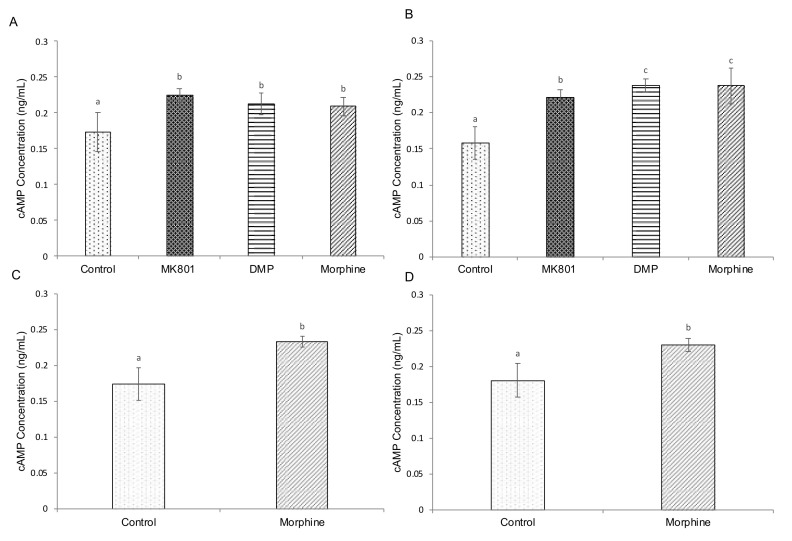
Effects of (**A**) short-term (F (3,32) = 22.035, *p* < 0.001) and (**B**) long-term (F (3,32) = 108.164, *p* < 0.001) exposure to MK801, DMP and morphine, and (**C**) short-term (*t* (16) = 12.465, *p* = 0.003) (**D**) and long-term (*t* (16) = 9.964, *p* = 0.029) morphine withdrawal on whole-body cAMP concentration (ng/mL) in zebrafish larvae. Data expressed as mean ± SEM, *n* = 50 of 3 independent experiments. Different lowercase letters indicate statistically significant values, *p* < 0.05. cAMP: cyclic adenosine monophosphate, DMP: domperidone.

**Figure 4 ijms-23-04715-f004:**
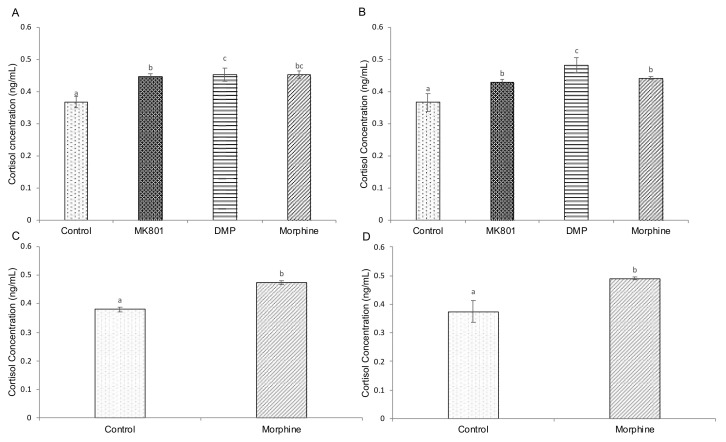
Effects of (**A**) short-term (F (3,32) = 22.035, *p* < 0.001) and (**B**) long-term (F (3,32) = 108.164, *p* < 0.001) exposure to MK801, DMP and morphine, and (**C**) short-term (*t* (16) = 12.465, *p* = 0.003) (**D**) and long-term (*t* (16) = 9.964, *p* = 0.029) morphine withdrawal on whole-body cAMP concentration (ng/mL) in zebrafish larvae. Data expressed as mean ± SEM, *n* = 50 of 3 independent experiments. Different lowercase letters indicate statistically significant values, *p* < 0.05. DMP: domperidone.

**Figure 5 ijms-23-04715-f005:**
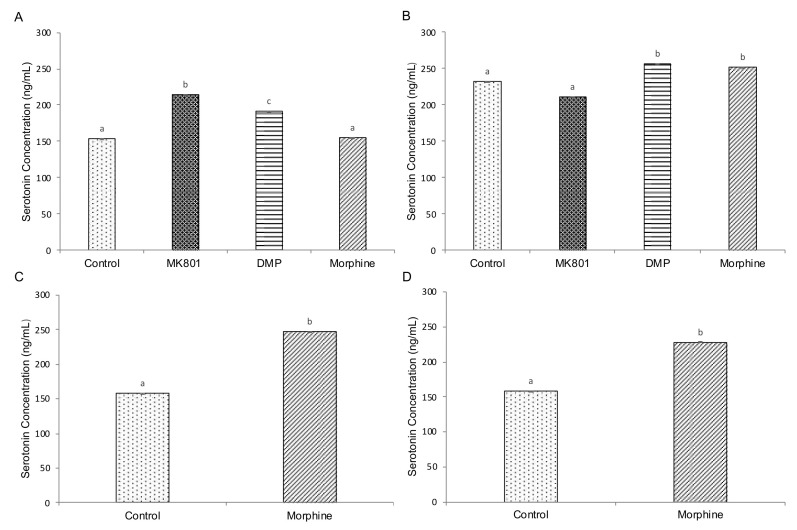
Effects of (**A**) short-term (F (3,32) = 16.064, *p* < 0.001) and (**B**) long-term (F (3,32) = 11.777, *p* < 0.001) exposure to MK801, DMP and morphine, and (**C**) short-term (*t* (16) = 4.531, *p* = 0.001) and (**D**) long-term (*t* (16) = 2.032, *p* = 0.019) morphine withdrawal on whole-body serotonin concentration in zebrafish larvae. Data expressed as mean ± SEM, *n* = 50 of 3 independent experiments. Different lowercase letters indicate statistically significant values, *p* < 0.05. DMP: domperidone.

**Figure 6 ijms-23-04715-f006:**
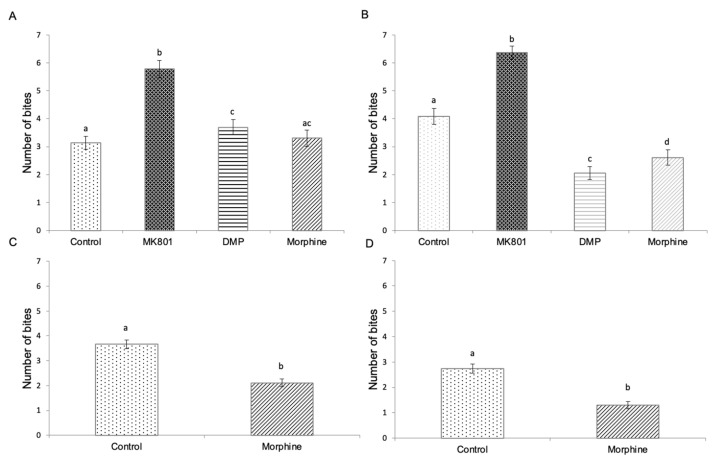
Mirror biting frequency of zebrafish larvae following (**A**) short-term (F (3,140) = 19.490, *p* = 0.001) and (**B**) long-term (F (3,140) = 56.224, *p* = 0.001) exposure to MK801, DMP and morphine, and (**C**) short-term morphine exposure followed by 5-day withdrawal (*t* (70) = 5.278, *p* = 0.001) and (**D**) long-term morphine exposure followed by 5-day withdrawal (*t* (70) = 8.830, *p* = 0.005). Data expressed as mean ± SEM, *n* = 36 of 3 independent experiments. Different lowercase letters indicate statistically significant values, *p* < 0.05. DMP: domperidone.

## Data Availability

Not applicable.

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
