# Peer review of "Receptor-Mediated AKT/PI3K Signalling and Behavioural Alterations in Zebrafish Larvae Reveal Association between Schizophrenia and Opioid Use Disorder"

_ijms, 2022, doi:10.3390/ijms23094715_

Round 1

Reviewer 1 Report

The manuscript from Thiagarajan et al. utilizes zebrafish models to study dopamine signaling modifying drugs such as morphine. Key aspect is studying responses to drug exposure and withdrawal. The topic is of high importance for fields of schizophrenia and drug abuse. Overall, the manuscript is well written and sound, although I have some recommendations for improvement of the manuscript.

General comments:

Results: For each section (2.1. , 2.2., 2.3, 2.4) it would be helpful to start with a introductory sentence which would explain why these experiments are done. Now, the text start with results, straight away, and readers who are not familiar with the topic might have some difficulties to follow the reasoning.

Figure 1: The figure 1 was distributed along several pages, and therefore a bit difficult to follow. Figure 1 could be potentially reorganized to improve readability.

Figure 6. This figure is quite illustrative and helps to understand the experiments. Please consider, if inclusion of these panels in the results section or other figures would improve the clarity of the manuscript.

Specific comments:

Lines 114-115: please specify, which phenotypic abnormalities (dead, malformation?) were measured.

Line 120-121: The sentence makes no sense, please rewrite.

Methods: Please add a statement if experiments were randomized or blinded.

Reviewer 2 Report

Comments

1. Research design does not answer the research question/hypothesis. Research design is not rigorous and clear.

2. Data for Optimization of drug dosages are sketchy, the number of animals is small (n=5), there are no statistical data to support these dose levels . The authors should add data for this section (table A1 is not enough).

3. There is a lack of evidence to prove that domperidon and morphine at doses 3.13 and 0.8 can induce Schizophrenia and Opioid Use Disorder. So the results of the manuscript are not significant and do not support the research hypothesis.

4. The use of only one dose level of domperidon and Morphine makes interpreting the results difficult. Authors should design at least 3 dose levels to see a relationship between dose and effect.

5. Using MK801 as positive control is inappropriate. It is not specific for both Schizophrenia and Opioid Use Disorder. We can not conclude that domperidon or morphine can induce Schizophrenia based on comparison with MK801.

6. The authors should add information for primers used in the study.

Some minor errors:

  • The line 75- 77 were repeated 2 times. The same content was shown in lines 78 – 80.
  • Genes should be in lowercase and italic in Figure 1

Round 2

Reviewer 2 Report

The manuscript improved and corrected so it can be accepted to be published.